# Application of Compressed Sensing Based on Adaptive Dynamic Mode Decomposition in Signal Transmission and Fault Extraction of Bearing Signal

Zhixin Cai [1,2], Zhang Dang [1,2,3,*], Ming Wen [1,2], Yong Lv [1,2] and Haochun Duan [1,2]

1   Key Laboratory of Metallurgical Equipment and Control Technology, Ministry of Education,
    Wuhan University of Science and Technology, Wuhan 430081, China; caizhixin@wust.edu.cn (Z.C.);
    wm120397@163.com (M.W.); lvyong@wust.edu.cn (Y.L.); dhc130514@163.com (H.D.)
2   Hubei Key Laboratory of Mechanical Transmission and Manufacturing Engineering,
    Wuhan University of Science and Technology, Wuhan 430081, China
3   National Demonstration Center for Experimental Mechanical Education,
    Wuhan University of Science and Technology, Wuhan 430081, China
*   Correspondence: dangzhang@wust.edu.cn; Tel./Fax: +86-027-6886-2212

**Abstract:** Bearings are widely used in mechanical equipment; nevertheless, potential dangers are also widespread, making bearing fault detection very important. For large equipment, the amount of collected signals tends to be huge, which challenges both signal transmission and storage. To solve this problem, compressed sensing (CS), based on specific dynamic modes of adaptive truncated rank dynamic mode decomposition (ADMD), is adopted to achieve the purpose of compressing and transmitting the signal, as well as extracting fault features. Firstly, this paper has proposed a new fitness function, which is called the synthetic envelope kurtosis characteristic energy difference ratio, and adopted the improved particle swarm optimization algorithm (IPSO) to select the best truncated rank adaptively. Then, the historical signal attempts to be decomposed into a series of temporal and spatial coherent modes, through ADMD, and those modes are filtered and cascaded into a highly self-adaptive dictionary, the DMD dictionary, which approximates the original signal with some error. Next, CS is employed to compress and reconstruct the signal, in order to reduce storage space and improve transmission efficiency. Finally, signals of high quality can be reconstructed through orthogonal matching pursuit (OMP) algorithm. Compared with traditional dictionaries, the DMD dictionary, based on the mode structure generated by ADMD decomposition, ass proposed in this paper, can better represent the original signal in the simulation signal and have good noise reduction performance. The correlation coefficient (CORR) between the reconstructed signal and noise signal is 0.8109, between the reconstructed signal and non-noise signal is 0.9278, and the root mean square error (RMSE) is 0.0659 and 0.0351, respectively. Compared with the traditional SVD and EMD denoising methods, ADMD-CS has better noise reduction performance. In this paper, the signal-to-noise ratio (SNR) is taken as the quantitative indicator of denoising performance. It is found that the SNR of simulation signal and experimental signal processed by ADMD-CS is higher than that of the traditional denoising methods, which is 0.3017 and 0.8407, respectively. The storage space of the signal is quite smaller than traditional methods, and the compression ratios (CR) of the simulation and experimental signals are 66.16% and 59.08%, respectively. In conclusion, ADMD-CS has a good application prospect in signal transmission, storage, and feature extraction.

**Keywords:** dynamic mode decomposition; compressed sensing; dictionary construction; sparse representation; denoising

## 1. Introduction

With the widespread application of bearings in manufacturing machinery and equipment, bearing faults may cause serious equipment and personnel safety issues. Therefore,

bearing fault issues have been attracting increasing attention. Generally, bearing condition detection includes two sequential processes of feature extraction and fault diagnosis [1]. Fault feature extraction is a reference to system health and decision-making, on which many researchers have done a lot of research [2,3]; thus, it plays an important role in system monitoring. For the monitoring system of large equipment, the collection of signals needs to obey the Nyquist sampling theorem, making the amount of signals collected huge [4] and challenging the transmission and storage equipment of the system. Meanwhile, due to the limitations of environmental factors, the signal obtained by the sensor needs to be transmitted and stored to the analysis system for further analysis, which requires the efficient and high-quality transmission of data. As an effective method, compressed sensing (CS) can effectively relieve the storage pressure of the system and reduce the cost of signal processing. Meanwhile, the large compression rate also makes CS more attractive [5].

CS is a theory for data compression and recovery, which is proposed by Donoho [6]. Generally, time of transmission will be greatly extended when the amount of data is large. According to the theory of CS, the original signal, which is sparse in a certain domain, often contains a lot of redundant data, which has little effect on the dynamic features of the original signal. Therefore, important data containing dynamic properties can be kept in the signal compression process, making the length of the compressed signal much shorter than the original signal. The advantage of CS is that the compressed signal not only reduces the requirements for hardware, but also improves the efficiency of data transmission.

In recent years, CS has been extensively used in many industries, such as medicine [7], imaging [8], and underwater communications [9]. In mechanical fault diagnosis, many scholars have proposed various methods based on CS theory [10,11]. In CS theory, the random Gaussian matrix is extensively adopted as a measurement matrix to satisfy restricted isometric property (RIP) condition to the greatest extent [12,13]. CS reconstruction algorithms, such as matching pursuit (MP) [14], basis pursuit (BP) [15], and iterative thresholding (IT) [16], have been developed and utilized by many scholars. Regarding dictionary selection, on the one hand, wavelet dictionary [17], Fourier dictionary [18], and discrete cosine dictionary [19] are traditional dictionaries. On the other hand, there are also dictionaries that are constructed by the different signal characteristics, such as Symlet dictionary and Daubechies dictionary [20]. Even so, it is still difficult for traditional dictionaries to represent noise signals sparsely. Therefore, a self-adaptive dictionary that can be constructed by the characteristics of the signal is needed. In the method of mining data characteristics, the modes of dynamic mode decomposition (DMD) contain fruitful dynamic information. Compared with the traditional dictionaries, the constructed DMD dictionary has the characteristics of the signal and is more self-adaptive. Meanwhile, it is theoretically considered that the DMD dictionary has a great potential to represent the original signal.

DMD [21] is an equation-free and data-driven frequency analysis method, based on singular value decomposition (SVD) and Koopman spectrum analysis, which combines the nonlinear dynamic systems and measured results of complex systems with the recognized method in dynamic system theory [22]. Compared with the application of traditional fault feature extraction methods in bearing signals [23,24], DMD has a good application prospect in bearing fault diagnosis. The DMD method decomposes the signal, which is rich in mechanical properties [25], into a series of single-frequency and non-orthogonal modes with inherent dynamic characteristics, the reconstructed signal formed by the filtered mode matrix and time matrix can describe the dynamic characteristics of the original signal well. Each decomposed mode corresponds to an eigenvalue, the real part represents the growth rate, and the imaginary part represents corresponding frequency.

It has been proven that CS can compress and reconstruct the signal perfectly and relieve the pressure of equipment transmission and storage, while DMD can decompose original signal and obtain fruitful temporal and spatial coherency. Therefore, DMD and CS can complement each other. This idea has been fully reflected in previous studies [26–28] and is called compressed dynamic mode decomposition (CDMD). CDMD considers two hypotheses,

namely known and unknown full-state data. If full-state data is known, the data can be compressed, based on CS, and then decomposed by DMD. Then, the compressed mode can be calculated. Finally, the full-state modes of data, obtained through combing the full-state data and compressed modes, are utilized to characterize the flow field.

This paper puts forward compressed sensing, based on adaptive dynamic mode decomposition (ADMD-CS), adopting the frame of CS, in which the DMD dictionary is based on the specific dynamic modes obtained from adaptive truncated rank dynamic mode decomposition (ADMD). The original signal can be compressed through the DMD dictionary constructed by ADMD and efficiently reconstructed, in order to reduce the storage space and improve transmission efficiency. Different from previous combination of CS and DMD [28], ADMD-CS focuses on the representation of the current signal via modes obtained from ADMD. By concatenating the dominant modes obtained from ADMD, as well as the modes that have the largest inner product with the fundamental frequency and multiple frequency, the DMD dictionary can be built. The best truncated rank is selected automatically by the improved particle swarm algorithm (IPSO), which can improve DMD decomposition accuracy effectively. Due to the fundamental and multiple frequency, which can be estimated by [29], the new dictionary is utilized to represent the original signal based on the fact that deviation exists between actual and ideal signals, and it is also used to extract the fault frequency under the noise background. The deviation is based on the power ratio of the low-rank matrix to the noisy signal [30].

The main structure of this article is as follows: Section 2 introduces CS, DMD theory, and the construction process of the DMD dictionary. IPSO and error indexes are also included. In Section 3, the validity of the DMD dictionary and ADMD-CS is proven by processing the simulation signal. In Section 4, the effectiveness of ADMD-CS in the experimental signal is verified. The last section draws a conclusion to this paper.

## 2. Methodologies

### 2.1. Compressed Sensing

#### 2.1.1. Compression Process

Supposed that the signal $Y \in R^{n \times 1}$, collected by the sensors, is a one-dimensional signal; simultaneously, $Y$ satisfies the condition that it is sparse in a certain domain, and there will be a certain set of column bases (atoms) $\mathbf{\Psi} = [\psi_1, \psi_2, \cdots, \psi_n]_{n \times n}$; the signal $Y$ can be represented by the coefficient vector $S = [s_1, s_2, \cdots, s_n]$ linearly:

$$Y = \mathbf{\Psi}S = \sum_{i=1}^{n} \psi_i \times s_i \tag{1}$$

where $S$ is sparse coefficient. If the number of non-zero coefficients in $S$ is $k$, the signal is $k$-sparse. The smaller the $k$ is, the sparser the signal ($k \ll n$), and the reconstructed signal will be better.

$S = [s_1, s_2, \cdots, s_n]$ is the projection of signal $Y$ on the column bases.

$$s_i = \langle Y', \psi_i \rangle \tag{2}$$

If redundant data in the original signal is abandoned, the measurement matrix $\mathbf{\Phi} \in R^{m \times n}$ ($m \ll n$) is necessary and projects the signal $Y$ on a low-dimensional space to generate the signal $Y_m$, namely to sample and encode.

$$Y_m = \mathbf{\Phi}Y = \mathbf{\Phi}\mathbf{\Psi}S = \mathbf{A}S \tag{3}$$

where $Y_m \in R^{m \times 1}$ is the compressed signal. $\mathbf{A} = \mathbf{\Phi}\mathbf{\Psi}$ is called the sensing matrix.

Since the original signal $Y$ is compressed from $n$ to $m$ and $m \ll n$, the entire compression process can be regarded as a process of re-sampling or projection. The compressed signal $Y_m$ contains the main dynamic features of the original signal.

It should be noted that the measurement matrix is not randomly selected, the RIP condition must be satisfied [11], which can be expressed as:

$$(1 - \varepsilon)\|v\|_2^2 \leq \|\Phi v\|_2^2 \leq (1 + \varepsilon)\|v\|_2^2 \tag{4}$$

where $\varepsilon \in (0, 1)$, $v$ is a vector with $k$ sparsity, and $\|\cdot\|_p$ represents the $p$ norm.

However, it is a complicated process to verify whether the RIP condition is satisfied. Donoho [4] and Candes [11] point out that, if $\Phi$ is a random Gaussian matrix, the matrix probably satisfies the RIP condition. In addition, Baraniuk R [31] also gives an equivalent formula that satisfies the RIP condition: if $\Phi$ is incoherent to $\Psi$, $A$ satisfies the RIP condition with a high probability. The degree of incoherence is as follows:

$$\mu(\Phi, \Psi) = \sqrt{N} \cdot \max_{1 \leq l,j \leq N} |\langle \phi_l, \psi_j \rangle| \tag{5}$$

where $N$ is the row number of the measurement matrix, $\phi_l$ is the $l$-th row of the measurement matrix, and $\psi_j$ is the $j$-th column of $\Psi$.

In order to reconstruct the original signal, $m$ needs to satisfy $m \geq 2ck\mu \log(N)$ (where $c$ is a small constant [32]). The smaller $\mu$ is, the stronger the incoherence, and the original signal can be reconstructed with a higher probability. Meanwhile, Steven LB et al. have proven that Bernoulli random measurement matrices and Gaussian random measurement matrices can satisfy the RIP with high probability [28].

### 2.1.2. Reconstruction Process

The reconstruction algorithm of compressed signals is very important to CS theory. The reconstruction of the original signal can be regarded as the process of solving $Y_m = AS$, given $Y_m$, $S$ is the recovered sparse coefficient, where $A \in R^{m \times n}$, $m < n$. The equation system is an underdetermined equation system, and only sparsity $k$ is given. Therefore, recovering $S$ is a problem of finding the optimal $l_0$-norm, namely minimizing the number of non-zero items in $S$, which is a hard problem. If the $l_0$-norm problem transforms into the $l_1$-norm problem, it will turn out to be a convex problem, which can be solved by convex optimization [33]. The process of recovering $S$ is as follows:

$$\widetilde{y} = \mathrm{argmin}\|S\|_1 \, s.t. \|Y - AS\|_2 \leq \varepsilon \tag{6}$$

where $\|\cdot\|_p$ represents the $p$-norm, and $\varepsilon$ represents the residual.

The most common convex optimization algorithm is basic pursuit (BP), which uses the $l_1$-norm to solve the optimization problem by using linear programming methods, and the original signal $Y$ can be recovered through the dictionary.

The emphases and difficulties of CS mainly depend on the selection of the measurement matrix $\Phi$, (orthogonal) base $\Psi$, and reconstruction algorithm. Commonly used reconstruction algorithms include $l_1$-norm [14], matching pursuit algorithm [15], and iterative threshold [16]. The process of CS is shown in Figure 1.

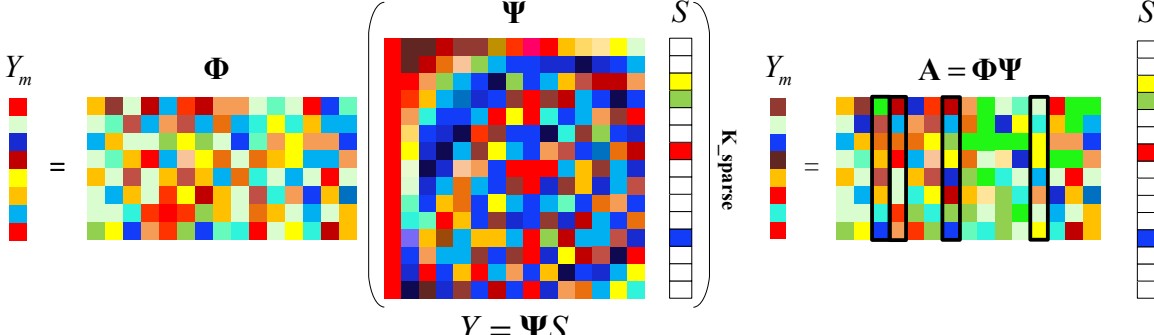

**Figure 1.** Compressed sensing process.

### 2.2. Dynamic Mode Decomposition

Suppose that the signal collected by sensors with equal intervals ($\Delta t = t_{i+1} - t_i$) is $X$, which can be expressed as $X = [X_1, X_2, \cdots X_N]$, $X_i \in R^{m \times 1}$, and the Hankel matrix can be obtained by a sliding window with fixed length over the corresponding vector $X$:

$$X = \begin{bmatrix} x_1 & x_2 & \ldots & x_n \\ x_2 & x_3 & \ldots & x_{n+1} \\ \ldots & \ldots & \ldots & \ldots \\ x_m & x_{m+1} & \ldots & x_{m+n+1} \end{bmatrix} = \begin{bmatrix} X_1 & X_2 & \ldots & X_n \end{bmatrix}, \; X \in R^{m \times n} \tag{7}$$

where $N = m + 1 - n$, and the Hankel matrix can be decomposed into two shift-stack matrices, $X_t$ and $X_{t+1}$, in the form of Formula (8), which are recognized as two continuous sequence snapshots.

$$X_t = \begin{bmatrix} X_1 & X_2 & \ldots & X_{n-1} \end{bmatrix}, \; X_{t+1} = \begin{bmatrix} X_2 & X_3 & \ldots & X_n \end{bmatrix} \tag{8}$$

Assuming that the two matrices satisfy the best mapping relationship, matrix **A** is utilized to express the dynamic characteristics between the two matrices, which is as follows:

$$X_{t+1} = \mathbf{A} X_t \tag{9}$$

where **A** contains abundant dynamic information, and the DMD algorithm needs to find an optimal low-rank matrix of **A** through the POD algorithm projection. At the same time, $X_t$ is decomposed by SVD:

$$\mathbf{A} = \mathbf{U} \hat{\mathbf{A}} \mathbf{U}^T, \; X_t = \mathbf{U} \mathbf{\Delta} \mathbf{V}^T \tag{10}$$

where $\mathbf{U} \in R^{m \times r}$ and $\mathbf{V}^T \in R^{r \times m}$ are the left and right eigenvector matrix, respectively. $\mathbf{U}\mathbf{U}^* = \mathbf{I}$, $\mathbf{V}\mathbf{V}^T = \mathbf{I}$, and $\mathbf{\Delta}$ are the non-zero diagonal matrices.

The best solution, $\hat{\mathbf{A}}$, is obtained by minimizing the F-norm error, $E$, between $X_{t+1}$ and $\mathbf{A}X_t$.

$$E = \text{argmin} \left\| X_{t+1} - \mathbf{U} \hat{\mathbf{A}} \mathbf{\Delta} \mathbf{V}^T \right\|_F \tag{11}$$

Then, the best approximate solution can be obtained from Formula (11):

$$\hat{\mathbf{A}} = \mathbf{U}^T X_{t+1} \mathbf{V} \mathbf{\Delta}^{-1} \tag{12}$$

$\hat{\mathbf{A}}$ is the low-rank expression of **A**, after similarity transformation; $\hat{\mathbf{A}}$ and **A** are approximately equal in dynamics. Then, apply eigenvalue decomposition to $\hat{\mathbf{A}}$:

$$\hat{\mathbf{A}} = \mathbf{W} \mathbf{\Lambda} \mathbf{W}^{-1} \tag{13}$$

where $\mathbf{W} = [\omega_1, \omega_2, \ldots, \omega_r] \in R^{m \times r}$ is the eigenvector of $\hat{\mathbf{A}}$, and $\mathbf{\Lambda} = [\lambda_1, \lambda_2 \ldots \lambda_r]$ is the eigenvalue matrix. Thus, the $i$-order mode of $\hat{\mathbf{A}}$ can be obtained by the eigenvector.

$$\phi_i^m = X_{t+1} \mathbf{V}_i \mathbf{\Sigma}_i^{-1} \mathbf{W}_i \tag{14}$$

where $\phi_i^m$ is the standard DMD mode.

For the convenience of expression, let $\omega_i = \ln(\lambda_i)/\Delta t$, and the approximate reconstruction solution of the original signal, $X$, is as follows:

$$X_{DMD} = \phi_i^m \exp(\omega_i \Delta t) b_i = \mathbf{\Phi}^m \exp(\mathbf{\Omega} t) b, \; X_{DMD} \in R^{m \times n} \tag{15}$$

where $\mathbf{\Phi}^m$ is the matrix containing the DMD modes, $\mathbf{\Omega} = \text{diag}(\lambda_i)$ is a diagonal matrix, whose elements are the eigenvalues of the similarity matrix $\hat{\mathbf{A}}$, $b = (\mathbf{\Phi}^m)^\Gamma X_t$ is a vector containing the magnitude of each mode, and $\Gamma$ represents the Moore-Penrose pseudo-inverse.

### 2.3. The Improved Particle Swarm Optimization Algorithm

Although the original signal can be reconstructed well via DMD, the selection of truncated rank $r$ has always been a key issue. The size of $r$ will determine the number of modes. If $r$ is too large, some unnecessary modes containing much noise may be used for reconstruction, thus causing low reconstruction quality. If $r$ is too small, the reconstructed signal will lose useful components. Hence, the improved particle swarm algorithm (IPSO) is proposed to adaptively select $r$.

PSO is a swarm intelligence algorithm proposed by Kennedy [34]. Through sharing information among groups, it will sort the movements of the whole group in the solution space for the optimal solution. Each particle has its own corresponding position and velocity vectors, and their fitness can be calculated through the fitness function of the current position In each iteration, the particles in the whole group will become the best among the whole population, leading to the occurrence of optimal solution in the total group [35]. Figure 2 exhibits the algorithm flow chart.

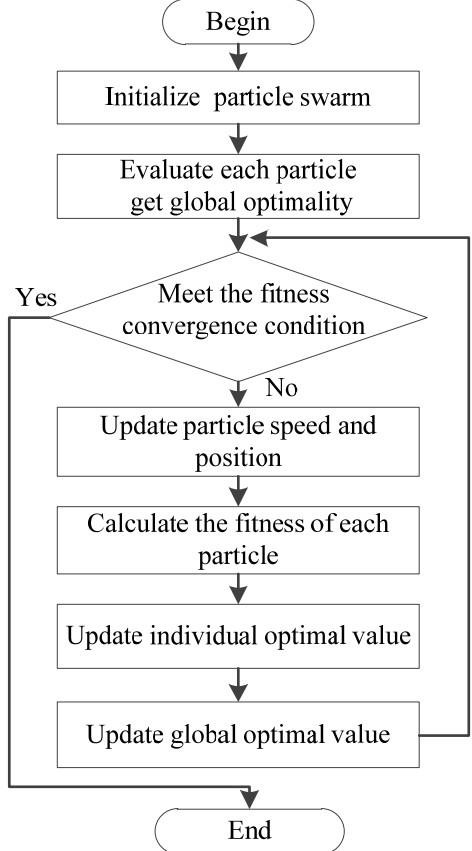

**Figure 2.** Flow chart of particle swarm optimization.

It is supposed that, if there is a D-dimensional target search space, where a whole group is composed of N particles, the position ($W_i$) and velocity ($V_i$) of the $i$-th particle are also D-dimensional vectors, $i = 1, 2, \cdots, N$.

$$W_i = (w_{i1}, w_{i2}, \cdots, w_{iD}) \tag{16}$$

$$V_i = (v_{i1}, v_{i2}, \cdots, v_{iD}) \tag{17}$$

The individual extreme value, denoted as $p_{best}$, refers to the optimal position searched by the *i*-th particle. The final optimal position of the whole particle swarm is the global extreme value, denoted as $g_{best}$.

$$p_{best} = (p_{i1}, p_{i1}, \cdots, p_{i1}) \tag{18}$$

$$g_{best} = (p_{g1}, p_{g1}, \cdots, p_{g1}) \tag{19}$$

The formulas of updated particle velocity and position are as follows.

$$v_{id} = \omega \times v_{id} + c_1 r_1 (p_{id} - x_{id}) + c_2 r_2 \left(p_{gd} - x_{id}\right) \tag{20}$$

$$x_{id} = x_{id} + v_{id} \tag{21}$$

where $c_1$ and $c_2$ are learning factors, and $r_1$ and $r_2$ are uniform random numbers within the range of [0, 1].

In PSO, the parameter inertia weight ($\omega$) is the most important. With the value of $\omega$ increasing, the global search ability of the algorithm will be enhanced. With the value of $\omega$ decreasing, the local search ability of the algorithm will be improved [36]. When traditional PSO algorithm is employed, its results easily fall into local optima. Aimed at this problem, the nonlinear dynamic inertia weight is used to achieve the effect of the global search, instead of the local search [37]. The nonlinear dynamic inertia weight coefficient formula is as follows:

$$\omega = \begin{cases} \omega_{\min} - \frac{(\omega_{\max} - \omega_{\min}) \times (f - f_{\min})}{f_{avg} - f_{\min}}, f \le f_{avg} \\ \omega_{\max}, f > f_{avg} \end{cases} \tag{22}$$

where $f$ represents the real-time objective function value of the particle, and $f_{avg}$ and $f_{\min}$ represent the average and minimum target values of all current particles, respectively. It can be seen from the above formula that the inertia weight changes with the value of particle objective function.

The basic steps of the IPSO are as follows:

(1). Initialize the position and velocity of each particle in the population randomly.
(2). Evaluate the fitness of each particle. The position and fitness of all particles will be stored in the individual extreme value ($p_{best}$), and the best position and fitness value in all $p_{best}$ will be put into the global extreme value $g_{best}$.
(3). Update the particle displacement and velocity in Equations (20) and (21).
(4). Update the weight in Equation (22).
(5). Compare the fitness value of each particle with its best position and take the current fitness value as the best position of the particles if they are close. Compare current $p_{best}$ and $g_{best}$ to update $g_{best}$.
(6). When the termination requirement is met, the search will stop, and results will output. Otherwise, it will go back to step (3), and go on.

When IPSO is used for optimization, it is necessary to define an objective function to measure how great the optimized parameters are. Then, the optimal solution of the whole particle swarm will be determined through the value of objective function.

Characteristic energy can be used to measure periodic shocks [38]. Meanwhile, the energy value of the characteristic frequency is larger than that of other noise components. In order to reserve energy value after each decomposition, the characteristic energy difference is adopted to better represent the decomposition effect. At the same time, when a periodic pulse component appears in the time domain, obvious fault frequency and its multiplicative components will emerge in the envelope spectrum [39]. Synthetic kurtosis is a new index, composed of envelope spectrum amplitude and kurtosis. Among them, the kurtosis index, a parameter describing waveform kurtosis is used to evaluate fault impact, due to its sensitivity to impact signals [40]. Additionally, the kurtosis of the envelope spectrum can

be employed to evaluate the cyclic stability of fault impact [41]. The combination makes up the limitation that the kurtosis index cannot be used to describe periodic stability [42]:

$$ESK = \frac{\sum_{j=1}^{p} \left| \overline{SE(j)} \right|^4}{\left( \sum_{j=1}^{p} \left| \overline{SE(j)} \right|^2 \right)^2} \tag{23}$$

$$EK = ESK \times ku \tag{24}$$

where *ku*, *EK*, and *ESK* represent kurtosis, synthetic kurtosis, and envelope spectrum amplitude kurtosis, respectively. *SE* is the envelope spectrum of signals, and *p* represents sampling points of the envelope spectrum. When the periodic impact is obvious in the decomposition result, the synthetic kurtosis value is large.

Therefore, this paper proposes a synthetic envelope kurtosis characteristic energy difference ratio as an adaptive fitness index:

$$Function = \frac{\text{EDV}}{EK \times E} \tag{25}$$

where *Function* refers to the fitness function, EDV represents the characteristic energy difference, *EK* represents the synthetic kurtosis, and *E* represents the characteristic energy. When periodic shock is not obvious, *Function* is large; otherwise, *Function* is small. Therefore, this paper can reflect on the quality of the decomposition results by monitoring the changing trend of the fitness function. When the fitness function approaches the minimum value, decomposition results are optimal, and the best truncated rank is obtained.

*2.4. Mode Selection and Dictionary Construction*

$\mathbf{\Phi}^*$ obtained by DMD is a matrix containing a series of temporal and spatial modes, which can reflect the dynamic characteristics of the original dynamic system. First of all, the best truncated rank obtained by IPSO is adopted to pick out the main modes, and manually truncated rank is also utilized to generate a mode matrix. Then, select the corresponding modes of the fault frequency estimated by the formula in [29]. At last, concatenate the selected modes with the modes selected by the best truncated rank to form a DMD dictionary; the purpose is to keep the useful components of the signal (as much as possible) and reduce the interference of noise. It is noted that the DMD modes appear in the form of conjugate, and sidebands appear due to the modulation of rotation frequency. Hence, fault frequency will be surrounded by two peaks in the process of sifting, and two modes that have a maximum inner product with the fault and multiple frequencies will be selected. The sparse coefficients of the formed dictionary tend to be unique, and the main components of the signal can be retained. The atom filtered by the inner product can be expressed as:

$$\mathbf{\Phi}^* = \left\{ \phi_{i_1 - i_2} \right\}, \ s.t. |\langle \phi_i \cdot \phi_b \rangle| = \max |\langle \phi_j \cdot \phi_b \rangle| \big|_{1 \sim 2}, \ j \in 1, 2, \dots, r \tag{26}$$

where $\phi_b$ is the mode matrix, where the frequency of each mode corresponds to the fault or multiple frequencies, and $\phi_{i_1 - i_2}$ represents a mode owning largest inner product with two modes matrices.

Since the filtered modes contain plenty of noise, in order to improve the quality of modes and represent the original signal better, the power of the filtered modes will be enhanced, so that the original signal can be approached within a certain error. The approximation error is based on the power ratio of the low-rank matrix **X** and noisy matrix **Y** [30], and the average power of the power signal is expressed as:

$$P = \frac{1}{t_2 - t_1} \int_{t_1}^{t_2} x^2(t) dt \tag{27}$$

The mode dictionary generation process is shown in Figure 3.

**Figure 3.** Flow chart of the dictionary construction process.

*2.5. Error Measurement*

In order to measure the quality and error of the reconstructed signal, the correlation coefficient (CORR) and root mean square error (RMSE) are employed to calculate the deviation of the reconstructed signal from the original signal. Meanwhile, the compression rate (CR) is used as the signal compression index, and the signal-to-noise ratio (SNR) is employed to present a quantitative index to evaluate the noise reduction performance of different methods. The calculation formula is as follows:

$$CORR = \frac{\sum\limits_{i=1}^{N} (x_i - \overline{x})(y_i - \overline{y})}{\sqrt{\sum\limits_{i=1}^{N} (x_i - \overline{x})^2 (y_i - \overline{y})^2}} \tag{28}$$

where $\overline{x}$ and $\overline{y}$ are the average, and $x$ and $y$ are two different vectors.

$$RMSE = \sqrt{\frac{1}{N} \sum_{i=1}^{N} \left(y^{(i)} - \hat{y}^{(i)}\right)^2} \tag{29}$$

where $y^{(i)}$ is the original signal, and $\hat{y}^{(i)}$ is the reconstructed signal.

$$CR = \frac{P - Q}{P} \times 100\% \tag{30}$$

where $P$ is the original signal size, and $Q$ is the compressed signal size.

$$SNR = 20 \log \frac{\|x\|_2^2}{\|x - \hat{x}\|_2^2} \tag{31}$$

where $x$ is the original signal, and $\hat{x}$ is the reconstructed signal. In conclusion, the schematic diagram of the proposed method is shown in Figure 4.

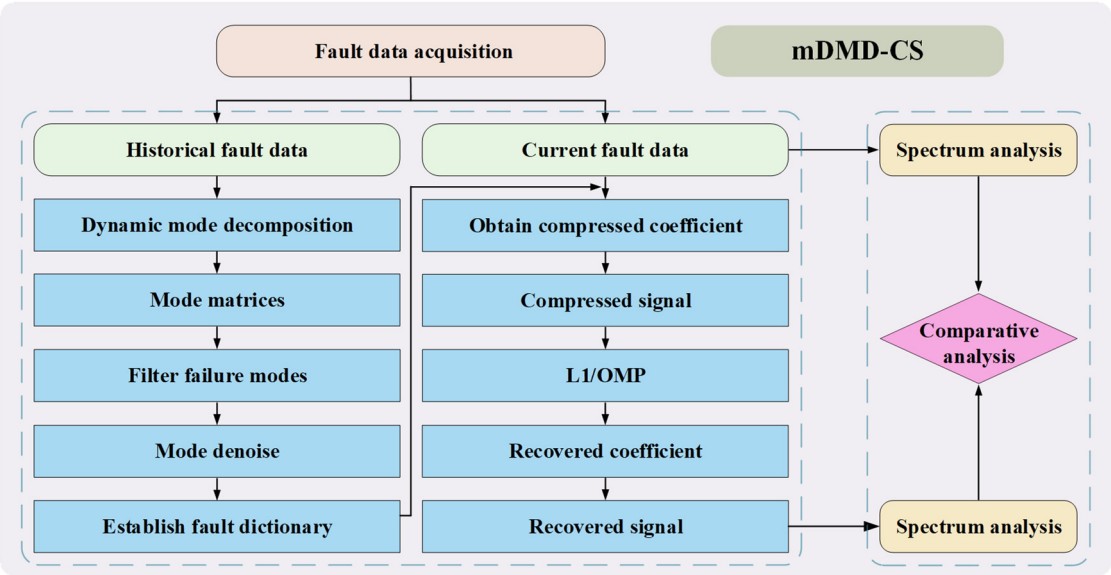

**Figure 4.** The schematic diagram of compressed sensing, based on adaptive dynamic mode decomposition.

## 3. Application of ADMD-CS in the Simulation Experiment

In this section, ADMD-CS is applied to the bearing simulation signal model established by Randall [43]. Due to the structural characteristics of the bearing, when the inner and outer rings of the bearing rotates relatively, the rolling elements will be subjected to different alternating loads, due to the different position. When the bearing faults, the bearing will generate an instantaneous impact signal and vibrate at its own inherent frequency.

Supposed that $dt$ is the sampling time interval, $S(t)$ is the natural oscillation frequency function of the bearing, $A_k$ is the amplitude of the $k$-th impulse response, and $N(t)$ is the background noise with zero mean, the model of the simulation signal can be expressed as:

$$
\begin{cases}
f(t) = \sum\limits_{k=1}^{M} A_k S(t - kT - \tau_k) + N(t) \\
A_k = a_k \cos(2\pi f_m t + \varphi_A) + c_A \\
S(t - kT - \tau_k) = \exp(-B(t - kT - \tau_k) \times \sin[2\pi f_n(t - kT - \tau_k) + \gamma]
\end{cases} \tag{32}
$$

where $a_k$ is the $k$-th impact energy, $\gamma$ and $\varphi_A$ are initial phases, $f_m$ is the modulation frequency, $B$ is the attenuation coefficient (relevant to the bearing system), $\tau_k$ is the lag time caused by fluctuations, and $c_A$ is a constant.

The parameter in Formula (32) is shown in Table 1. The fault frequencies of the inner ring, outer ring, rolling element, and cage are $f_i$ $f_o$, $f_b$, and $f_c$, respectively. Take the fault signal of the inner ring as an example. The sampling frequency is selected as $f_s = 6$ kHz, and the number of sampling points is $N = 5000$. At the same time, Gaussian white noise is taken into consideration, so as to make SNR reach 3dB. Haar dictionary,

Fourier dictionary, and discrete cosine dictionary are employed for comparison, in order to verify the superiority of the AMDM-CS. Firstly, the historical signal is generated through a simulation signal model and decomposed with ADMD; we take the order of Hankel matrix $s$ = 2048. Then, the best truncated rank is selected, $r$ = 32, by the IPSO, and the manual truncated rank is selected when the energy of the eigenvalue is up to 90% of the total energy of all eigenvalues. Next, a new mode matrix is generated, and modes corresponding to the fault and multiple frequencies are picked out, according to Formula (26). Finally, the first mode matrix is concatenated with filtered modes to form a DMD dictionary, where each column represents a mode (an atom). At last, we get a dictionary with 91 atoms. The current signal is generated when 3 dB noise is added to the historical signal. The simulation bearing fault signal is shown in Figure 5. The parameters of the bearing simulation signal are shown in Table 1.

**Table 1.** Parameters of bearing simulation signal.

| $a_k$ | $\gamma$ | $\varphi_A$ | $\tau_k$ | $c_A$ | $B$ | $f_i$ | $f_o$ | $f_b$ | $f_c$ |
|---|---|---|---|---|---|---|---|---|---|
| 4 | 0 | 0 | 0.02 | 1 | 800 | 150 | 180 | 170 | 25 |

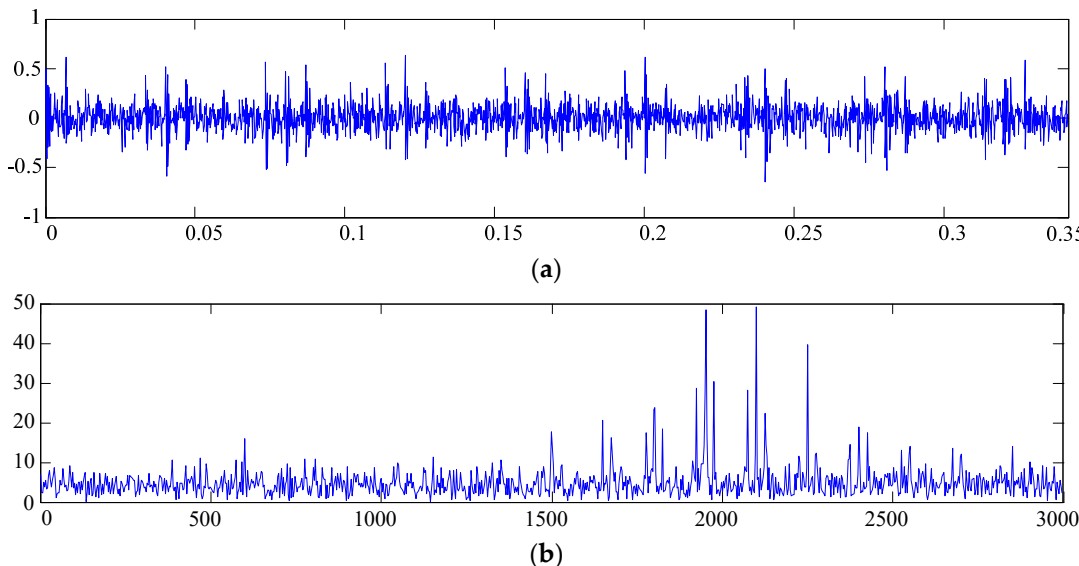

**Figure 5.** Time and frequency domain of simulation signal. (**a**) Time domain; (**b**) frequency domain.

To verify the sparsity of the DMD dictionary, it is compared with traditional dictionaries. The sparse coefficients of each dictionary are shown in Figure 6.

As shown in Figure 6, the sparsity of the sparse coefficients of each dictionary is relatively small. For the first three dictionaries, because they are stable and not designed for the signal, the dictionary owns a wide range of frequency. In addition, the signal contains noise, and their coefficients are not zero. The DMD dictionary is composed of the main modes containing the dynamic characteristics of the signal, so its coefficients are also non-zero. What is different is that its coefficient distribution is more regular, and there would be more elements close to zero than in other dictionaries. In other words, this coefficient vector is sparser. Furthermore, DMD modes are rich in signal characteristics, and they are more self-adaptable to signals. According to the estimation of the low-rank matrix in [30], the estimated power ratio of the noise-free signal to the noisy signal is 0.4511, and the signal represented by the DMD dictionary has a power ratio of 0.4594, with a difference of 0.0083. It can be considered that the DMD dictionary represents the original signal well.

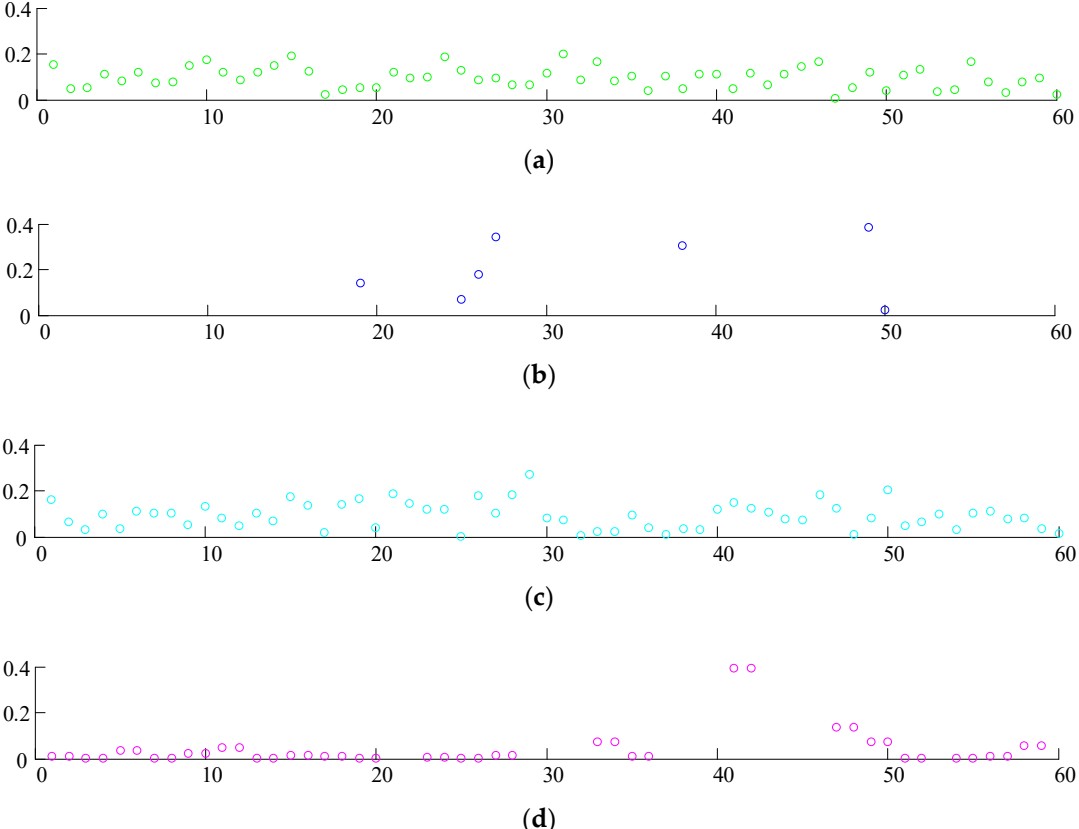

**Figure 6.** The sparse coefficient of each dictionary. (**a**) FFT coefficients; (**b**) Haar coefficients; (**c**) DCT coefficients; (**d**) DMD coefficients.

In actual application scenarios, the signal needs to be preprocessed and compressed in the sending module, then reconstructed in the receiving module. As mentioned above, there are many recovery algorithms for signals, typically including the L1, OMP, and IT algorithms. In order to select efficient and high-quality recovery algorithms, these three typical algorithms are compared in Figure 7. Then, the reconstruction time and quality of the recovery signal processed by the DMD dictionary are compared, and the quality is measured by RMSE and CORR, as proposed in Formulas (28) and (29). The reconstructed time and quality of each algorithm are shown in Figure 7.

As shown in Figure 7 and Table 2, among these three algorithms, the reconstructed time of the IT and OMP algorithms is much shorter than that of L1 algorithm, and the reconstructed time of the IT algorithm is almost equivalent to the OMP algorithm. In terms of reconstructed quality, the reconstructed quality of the IT algorithm is slightly lower than the other two algorithms (reconstruction quality of the OMP and L1 algorithms are too close, and their curves overlapped). However, the IT algorithm has higher signal requirements than other algorithms. Therefore, from the perspective of factors such as reconstruction time and quality, the OMP algorithm is the most suitable among three algorithms and has a wide range of applicability. It can be used as the reconstruction algorithm for experimental data.

**Table 2.** Average value of time and quality, reconstructed by different methods.

|        | L1    | OMP   | IT    |
|--------|-------|-------|-------|
| Time/s | 0.122 | 0.013 | 0.005 |
| RMSE   | 0.241 | 0.241 | 0.243 |

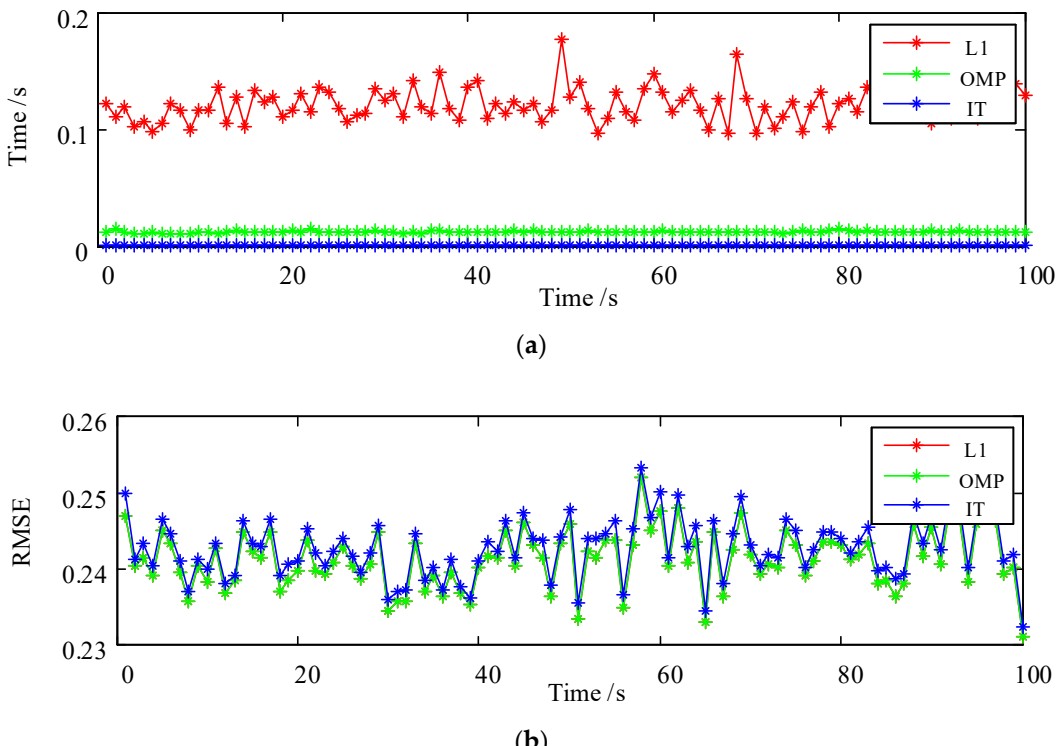

**Figure 7.** Reconstructed time and quality of different algorithms. (**a**) Reconstructed time; (**b**) reconstructed quality.

After the compression of the original signal, the simulation signal is reconstructed by the OMP algorithm; the reconstructed signals are shown in Figure 8. The CORR and RMSE of each reconstructed signal to the noisy signal and no-noise signal are shown in Table 3.

**Table 3.** Comparison of reconstructed signals with noise-free and noisy signals.

| DICTIONARY | | FFT | HAAR | DCT | DMD |
|---|---|---|---|---|---|
| Noisy signal | CORR | 0.3719 | 0.1184 | 0.4277 | 0.8109 |
| | RMSE | 0.1360 | 0.1857 | 0.1262 | 0.0659 |
| Noise-free signal | CORR | 0.4154 | 0.1358 | 0.5024 | 0.9278 |
| | RMSE | 0.1231 | 0.1747 | 0.1102 | 0.0351 |

In Figure 8, the reconstructed signal (e), via the DMD dictionary, is compared with (a), and most of the noise in (e) is removed; it is even similar to the noise-free signal (f). This is proven in Table 3 by the CORR and RMSE of the noise-free signal. Due to the wide frequency range of dictionary atoms and existence of noise, traditional dictionaries are not sparse enough in their respective domain; the noise is retained in the reconstruction process and even causes signal distortion. The DMD dictionary is composed of main modes containing signal characteristics, so useful components can be retained during the reconstruction process, and noises can be removed effectively, as well. As shown in Table 3, the correlation between the ADMD-CS reconstructed signal and noisy signal is better than other reconstructed signals. For the noiseless signal, the CORR and RMSE of ADMD-CS reconstructed signal are also better than other reconstructed signals. It could be conducted that ADMD-CS has a good denoising effect, and the DMD dictionary has good adaptability.

In order to judge the performance of the proposed method in denoising, ADMD-CS is compared with traditional denoising methods, such as SVD and EMD, in the frequency domain. Frequency domains of various methods are shown in Figure 9.

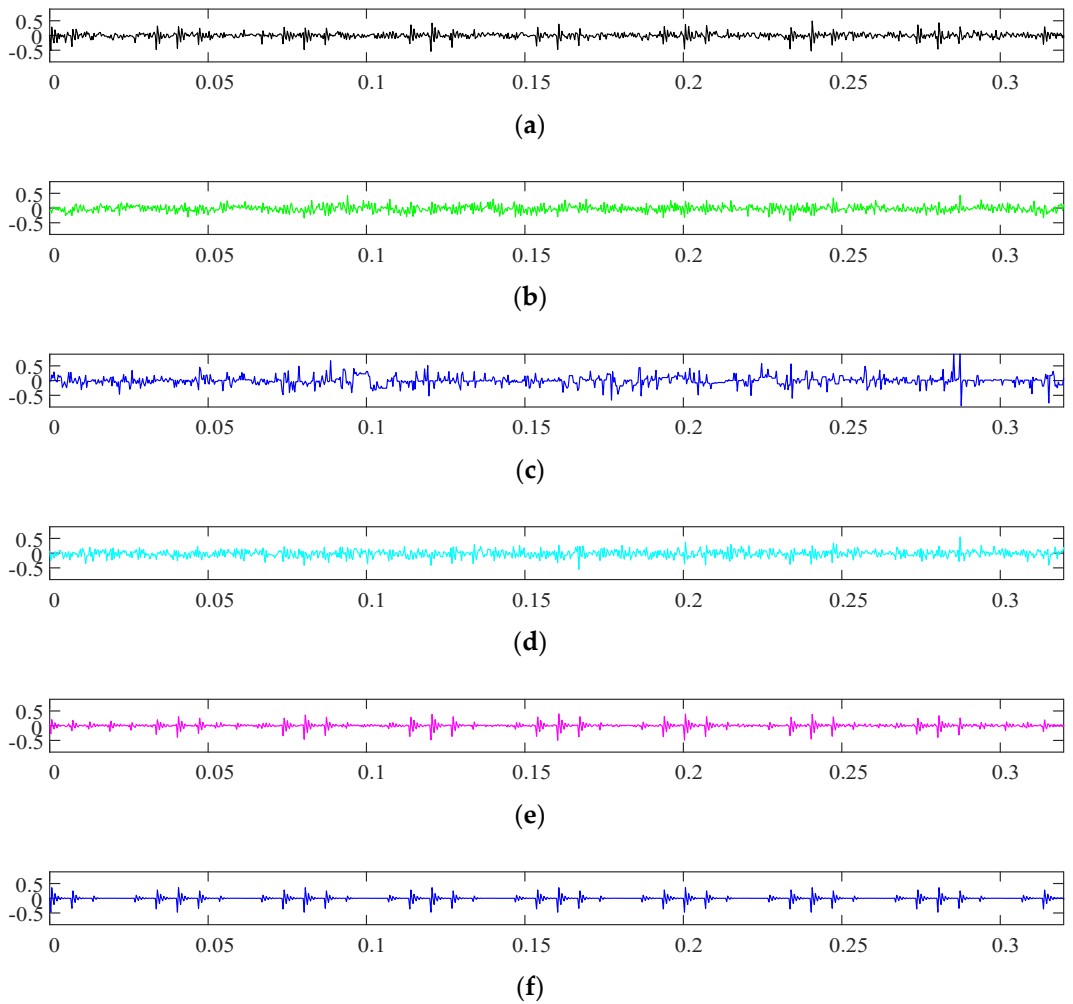

**Figure 8.** Simulation and reconstructed signals of each dictionary. (**a**) Simulation signal; (**b**) FFT; (**c**) Haar; (**d**) DCT; (**e**) DMD; and (**f**) noise-free signal.

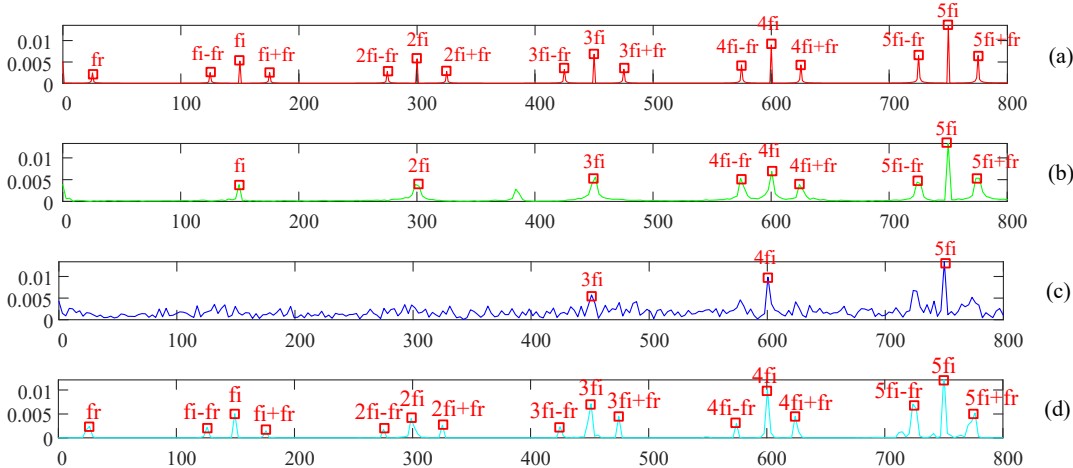

**Figure 9.** Comparison of various noise reduction methods. (**a**) Noise-free signal; (**b**) SVD; (**c**) EMD; (**d**) ADMD-CS.

As shown in Figure 9 and Table 4, compared with the traditional denoising methods, ADMD-CS has better effect than traditional methods in extracting the fault frequency by qualitative and quantitative indexes. The traditional SVD can remove plenty of noise; the low-energy fault frequencies submerged in the noise also can be removed, and the SNR

of the recovered signal is $-0.3188$. The EMD method decomposes the signal into some intrinsic mode functions (IMFs). Each IMF contains noise consequently, but only those noises in unnecessary IMF are removed during the reconstructed process. Those IMFs containing fault features, needed to reconstructed original signal, are still accompanied by noise. As a result, the fault frequency is submerged in the noise background and cannot be distinguished. SNR of the recovered signal by EMD is $-1.6664$. ADMD-CS uses the faulty atoms of the historical signal to extract useful components. The noise is less in each atom, compared to the original signal. Each atom corresponds to a single frequency; after the power is enhanced in frequency domain, the noise is further suppressed, so as to achieve the purpose of denoising and extracting the fault frequency. ADMD-CS can effectively remove the noise in the original signal; the obtained reconstructed signal is purer, and the characteristic and multiple frequencies can be clearly displayed. Meanwhile, its SNR is 0.3017, which is higher than that of SVD and EMD, consistent with the results shown in Figure 9, indicating that ADMD-CS has excellent performance in feature extraction and noise reduction.

**Table 4.** SNR of recovered and original signals, after processing the simulation signal with different methods.

| Methods | SNR |
|---------|-----|
| SVD | $-0.3188$ |
| EMD | $-1.6664$ |
| ADMD-CS | 0.3017 |

In actual application scenarios, CS is adopted as the transmission method, and the signal transmission cost and storage space are greatly reduced, compared with other traditional methods that need to collect and transmit a large amount of data. Signals reconstructed by traditional SVD, EMD, and ADMD-CS were saved as ".xls" format files, respectively; the file sizes were 99, 99, and 33.5 kb, respectively, and the size of the original signal was 99 kb, which is shown in Table 5. From the perspective of storage, the storage space of the signal processed by the traditional methods is almost unchanged, while the storage space of ADMD-CS is greatly reduced, with the reduction being about two-thirds of that of the traditional methods, and the CR is 66.16%. The effect of denoising is better than traditional methods, and it could be seen that the advantages of ADMD-CS are more obvious. Signal analysis and processing are often carried out after the signal transmits to the processing system. Compared to traditional fault feature extraction methods, ADMD-CS can be carried out in the signal sending module, which reduces the pressure on the signal receiving module.

**Table 5.** Storage space for simulation signal with different methods.

| Methods | Original Signal | SVD | EMD | ADMD-CS |
|---------|-----------------|-----|-----|---------|
| Storage space | 99 kb | 99 kb | 99 kb | 33.5 kb |

## 4. Application of Algorithm in Experimental Data

The actual fault signal is more complex than the simulation signal and easily interfered by noise during the transmission process. To verify the effectiveness of ADMD-CS in the experimental signal, the method is applied to the experimental signal data of rolling bearings, as published by Cincinnati University [44].

The structure of the Cincinnati University test bench is shown in Figure 10. Four Rexnord ZA-2115 double-row bearings were mounted on a 2000 rpm shaft, which was driven by an AC motor via sever rub belts. Each row of the bearing had $Z = 16$ rolling elements. The bearing pitch and rolling element diameter were $D = 28.15$ mm and $d = 3.31$ mm, respectively. The contact angle of bearings was $\alpha = 15.17°$. The vibration signal was collected by a high-sensitivity quartz ICP accelerometer, installed in the bearing box, and the signal was recorded via a NI DAQ Card 6062E.

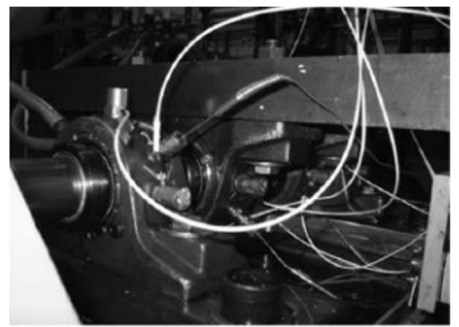 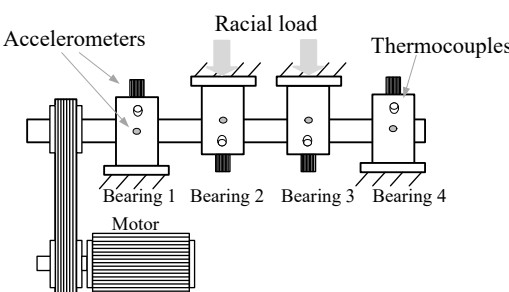

**Figure 10.** Cincinnati University bearing test bench.

The sixth channel of the first data set was employed, sampling frequency was set as $f_s$ = 20 kHz, and number of data points was set as $N$ = 20480. The rotation and fault frequencies were calculated as $f_r$ = 33.3 Hz and $f_i$ = 296.63 Hz. We take the order of the Hankel matrix: $s$ = 7000. The time domain and frequency domain of the signal are shown in Figure 11.

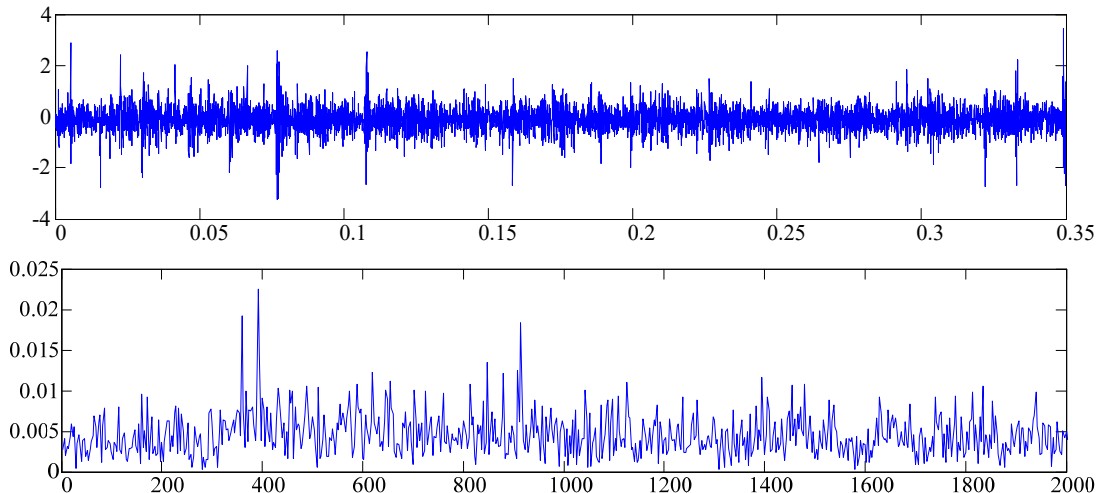

**Figure 11.** Time and frequency domains of the sixth channel signal.

As shown in Figure 11, only a few prominent spikes could be observed in the frequency domain. The fault frequency of the signal was submerged in noise and could not be distinguished; ADMD-CS was, thence, adopted to represent the original signal. The historical signal was decomposed by ADMD, and a new dictionary could be constructed via DMD modes—a total of 565 atoms were obtained. According to the literature [30], it is estimated that the measured signal power ratio is 0.3185, power ratio expressed by the DMD dictionary is 0.2589, difference is 0.0596, difference is small, and estimation is considered to be suitable. The signal is then reconstructed through the OMP algorithm.

To verify the effectiveness of the proposed method in noise reduction, the signal recovered by ADMD-CS was compared with the signal recovered by traditional SVD and EMD in the frequency domain. Additionally, the frequency domains of the different methods are shown in Figure 12.

Though the comparison of the three methods in Figure 12 and Table 6, it can be easily found that, though SVD can remove plenty of noise, the low-frequency fault component submerged in the noise was also removed as noise, and only the part of the fault frequency with strong energy was retained. While EMD decomposes the signal into several IMFs, noise exists in each IMF. When some IMFs containing fault features are used for reconstruction, the noise in each IMF is not removed, and it is even brought into the reconstructed signal, in turn, resulting a lot of noise in frequency domain, making it difficult to distinguish

and extract the fault frequency. ADMD-CS approximates the original signal by using the fault frequency modes obtained by the ADMD of the historical signal. The frequency range narrows when the dictionary is constructed, and the noise is further suppressed in the frequency domain because of enhanced power. The reconstructed signal is purer, making it possible to extract the fault frequency submerged in the noise. However, we should notice that, though the noise of the signal is suppressed, to a certain extent, and fault frequency is extracted successfully, there are still some unknown components in the frequency domain. To be honest, ADMD-CS has a better performance in fault extraction than the SVD and EMD methods. The SNR of the reconstructed signals processed by ADMD-CS was 0.8407, while that processed by SVD and EMD were 0.6193 and 0.2920, respectively, which also verifies that ADMD-CS has an excellent noise reduction performance. In addition, the traditional signal collection and transmission methods in actual application scenarios put great pressure on the collection and storage system. As can be seen from Table 7, ADMD-CS has better compression performance and CR is 59.08%. ADMD-CS compresses and reconstructs the signal based on CS, which not only alleviates the collection and storage space, but also performs preliminary signal processing, consequently reducing the pressure for subsequent further analysis; it has certain application potential in actual application scenarios.

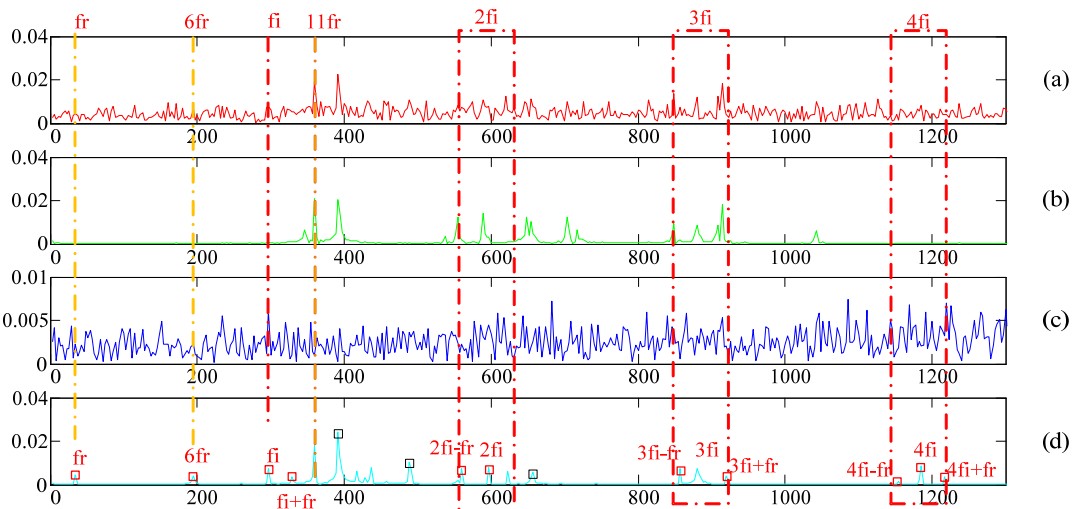

**Figure 12.** Frequency domains comparison of various algorithms. (**a**) Original signal; (**b**) SVD; (**c**) EMD; (**d**) ADMD-CS.

**Table 6.** SNR of the recovered and original signals, after processing experimental signal by different methods.

| Methods | SNR |
|---------|-----|
| SVD | 0.6193 |
| EMD | 0.2920 |
| ADMD-CS | 0.8407 |

**Table 7.** Storage space for the experimental signal by different methods.

| Methods | Original Signal | SVD | EMD | ADMD-CS |
|---------|-----------------|-----|-----|---------|
| Storage space | 50.1 kb | 50.1 kb | 50.1 kb | 20.5 kb |

## 5. Conclusions

As an equation-free and data-driven frequency analysis method, based on SVD and Koopman spectrum analysis, ADMD can describe dynamic features of the original system. Therefore, the modes obtained from ADMD can be constructed as a self-adaptive dictionary to represent the original signal, while selecting a suitable dictionary is a key problem of

CS theory. As a mutual complement, by combining these two features from DMD and CS, ADMD-CS is proposed, based on CS and ADMD.

In this paper, a self-adaptive dictionary, based on ADMD, is constructed; the original signal is reconstructed based on the actual situation, where there is an error between the ideal and actual signals. Firstly, the historical signal is decomposed by ADMD; then, the best truncated rank is selected through IPSO, and the energy modes are obtained. Simultaneously, the fault modes are picked out through inner product, with fundamental and multiple frequencies. Finally, the DMD dictionary is built up through concatenating the two mode matrices. Compared with traditional dictionaries, the DMD dictionary demonstrates a good effect on representing the original signal. The CORR between reconstructed signal and noise signal was 0.8109, and, between the reconstructed and non-noise signals, it was 0.9278; the RMSE was 0.0659 and 0.0351, respectively. Compared with the traditional denoising method of SVD and EMD, ADMD-CS has the better denoising effect. In this paper, SNR is used as a quantitative indicator of denoising performance. It was found that the SNR of the simulation and experimental signals processed by ADMD-CS was higher than that of traditional denoising methods, which were 0.3017 and 0.8407, respectively. The storage space of signal processed by ADMD-CS was quite smaller than the traditional methods, and the CR of the simulation and experimental signals were 66.16% and 59.08%, respectively. The main contributions of this paper are as follows:

(1). A new fitness function of the IPSO algorithm is defined, namely a synthetic envelope kurtosis characteristic energy difference ratio. Additionally, a better decomposition effect can be achieved by using optimized target parameters.

(2). A nonlinear dynamic inertia weight is used to optimize the traditional PSO algorithm, which is out of local search, and adaptively selects the truncated rank and threshold value to obtain a better decomposition effect and accurately extract fault features.

(3). Combined ADMD with CS to form a new method, namely ADMD-CS. It compresses and reconstructs the signal to achieve the goal of reducing storage space and improving transmission efficiency.

(4). Enhance the power of the mode in the frequency domain and use OMP to suppress the noise to obtain a better reconstructed signal.

Both DMD and CS have been widely used in their respective communities. This paper proposes the method of ADMD-CS for constructing an adaptive dictionary, signal transmission, and fault extraction. By applying ADMD-CS to simulation and experimental signals, the effect of representing the original signal and noise reduction are proved. The DMD dictionary is formed by energy and fault modes, and the truncated rank determines how many atoms there are in the energy mode matrix. The power of the modes is enhanced in the frequency domain to suppress noise. Though ADMD-CS outperforms traditional dictionaries and denoising methods, and noise has been better processed, there are still several unidentified components in the signal. In addition, the deviation of two power ratios in the experimental and simulated signals is a problem that should be paid attention to. SNR will influence this ratio; as a result, the performance of the extract fault features may be influenced. With different SNR in the signal, research on effect of this ratio and how to shrink this deviation will be our future work. Moreover, we will further analyze the signals collected in large equipment and further optimize them by using methods such as compression sampling matching tracking algorithm, so as to apply them to practical industrial production as soon as possible.

**Author Contributions:** Conceptualization, Z.D. and Z.C.; methodology, Z.D. and Z.C.; data acquisition, Z.C.; processing, Z.C.; writing manuscript, M.W.; writing–review, H.D. and Y.L.; editing, H.D. and Y.L.; supervision, H.D. and Y.L. All authors have read and agreed to the published version of the manuscript.

**Funding:** This work was supported by the National Natural Science Foundation of China (under grant no. 51575408), Innovation Group Project of Natural Science Foundation of Hubei province (under grant no. 2020CFA033), and Open Fund of Hubei Key Laboratory of Mechanical Trans-

**Institutional Review Board Statement:** Not applicable.

**Informed Consent Statement:** Not applicable.

**Data Availability Statement:** Research data are not shared.

**Conflicts of Interest:** The authors declare no conflict of interest.

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
