# Peer review of "Application of Compressed Sensing Based on Adaptive Dynamic Mode Decomposition in Signal Transmission and Fault Extraction of Bearing Signal"

_machines, doi:10.3390/machines10050353_

Round 1

Reviewer 1 Report

The authors proposed compressed sensing based on adaptive dynamic mode decomposition (ADMD-CS) for application in signal transmission, storage and fault extraction. Overall, they had a clear description of the proposed methods.

• Authors have presented the research design in a neat structured manner.

• Authors have compared three typical reconstruction algorithms i.e., Iterative thresholding (IT), OPM and L1 but they can compare performance of modified versions of OMP like orthogonal Multiple Matching Pursuit, Subspace Pursuits, Stage-wise OMP, Compressive Sampling Matching Pursuits algorithms too.

• Minor language correction is needed like dropping the word ‘many’ (5th word) before the word research on line 23.

Following are my concerns:

  1. I have main concerns about the motivation of the solution to the problems in signal transmission, storage, and fault extraction. Authors need a clear clarification of the problems they attempted to solve.
  2. I did not see quantitative evaluation results for the proposed methods in the abstract and conclusions.

Reviewer 2 Report

This paper proposes a new compressed sensing method based on adaptive dynamic mode decomposition (ADMD). The proposed method effectively reduces the storage space of data, and also shows better de-noising performance than the conventional methods. Though the authors presented the superiority of the proposed method using simulation and experimental data, there exist several critical issues that need to be revised.

  1. In the Introduction, the authors described overall compression sensing (CS) in detail. This can hinder the research motivation. Therefore, it is recommended to move these descriptions in Section 2.
  2. In Line 73 of the Introduction, the authors mentioned that the traditional dictionary is still difficult to represent noise signals sparsely. Here, the reviewer wonders why the dictionary should sparsely represent noise signals, which can derive the research motivation of this study.
  3. It is recommended to express the meaning of the colors in Figure 1.
  4. Basically, the proposed ADMD is a data-driven method. Then, how can the proposed method extract the fault frequency that is physical property? The authors only showed the results of the fault frequency extraction without the fundamental description of the proposed method.
  5. What is the difference between the DMD and the proposed ADMD? In Sections 3 and 4, it is recommended to conduct comparative studies using the exiting DMD.
  6. The reviewer found that the motivation of this study is originated from the large equipment. However, the authors only validated the proposed method using the small-scale testbed. Please address this issue.
  7. It is recommended to compare the performance of the noise reduction in a quantitative manner.

Round 2

Reviewer 1 Report

The authors revised the paper based on my review comments

Reviewer 2 Report

Accept it as it is.